# Exploration of a miRNA-mRNA network shared between acute pancreatitis and Epstein-Barr virus infection by integrated bioinformatics analysis

**Xing Wei**[1], **Zhen Weng**[2,3], **Xia Xu**[4]*, **Jian Yao**[1]*

**1** Department of Infectious Disease, The Nantong First People's Hospital and The Affiliated Hospital 2 of Nantong University, Nantong, China, **2** MOE Engineering Center of Hematological Disease, Soochow University, Suzhou, China, **3** Cyrus Tang Hematology Center, Collaborative Innovation Center of Hematology, Soochow University, Suzhou, China, **4** Department of Gastroenterology, The Second People's Hospital of Nantong and The Affiliated Rehabilitation Hospital of Nantong University, Nantong, China

☯ These authors contributed equally to this work.
\* nteyxuxia@163.com (XX); yaojian1977@163.com (JY)

**Data Availability Statement:** All datasets are available from the GEO database (GSE194331, GSE42455, GSE45918 and GSE109220). All

## Abstract

Acute pancreatitis (AP) stands out as a primary cause of hospitalization within gastrointestinal ailments, attributed to diverse factors, including Epstein-Barr virus (EBV) infection. Nevertheless, the common miRNAs and genes shared between AP and EBV infection remain unclear. In the present study, four datasets GSE194331, GSE42455, GSE45918 and GSE109220 were selected and downloaded from the Gene Expression Omnibus (GEO) database. Differential expression analysis was performed to screen for differentially expressed genes (DEGs) and differentially expressed miRNAs (DEMs). Target genes of overlapping DEMs were predicted, and intersections with overlapping DEGs were used to construct a miRNA-mRNA network. In addition, the enrichment analysis, drug prediction, diagnostic accuracy assessment, competitive endogenous RNA (ceRNA) network construction, transcription factor (TF)-miRNA-mRNA network construction, and immune cell infiltration analysis were also carried out. We found a total of 111 genes and 8 miRNAs shared between AP and EBV infection. A miRNA-mRNA network was constructed, which comprised 5 miRNAs and 10 genes exhibiting robust diagnostic performance. Histone deacetylase (HDAC) inhibitor was identified as a novel therapeutic intervention from drug prediction analysis. The results of immune cell infiltration analysis revealed that a consistent and significant difference could be found on activated B cell in AP and EBV-infected individuals in comparison to the controls. Taken together, our work, for the first time, revealed a miRNA-mRNA network shared between AP and EBV infection, thereby enriching a deeper comprehension of the intricate molecular mechanisms and potential therapeutic targets entwined in these two pathological conditions.

relevant data are within the paper and its supporting information files.

**Funding:** This study was supported by the National Natural Science Foundation of China [81700129] and the Translational Research Grant of NCRCH [2020WSA01].

**Competing interests:** The authors have declared that no competing interests exist.

## Introduction

Acute pancreatitis (AP) is a sudden inflammation of the pancreas, inducing intense abdominal pain and potentially life-threatening complications [1]. The etiology of AP encompasses various factors, including cholelithiasis, ethanol, hypertriglyceridemia, and even viral infections [2]. Notably, clinical evidence underscores a robust correlation between Epstein-Barr virus (EBV) infection and AP development [3–5]. In addition, approximately 90% of infectious mononucleosis (IM) cases attributed to EBV infection, which is characterized by swollen lymph nodes, fever, sore throat, and often extreme fatigue [6]. Cases of IM complicated with AP have also been reported, some of which have been clearly related to EBV infection [7–10]. The pancreas is likely one of the target organs affected by the immune response during infection, which leads to the occurrence of pancreatitis. Despite these association, the underlying common miRNAs, genes and molecular mechanisms shared between AP and EBV infection are intricately complex, and remain insufficiently elucidated to date.

Recent strides in high-throughput transcriptomic methodologies have paved the way for evaluating alterations in multiple genes or miRNAs expression levels concurrently, offering a panoramic insight into differentially expressed genes (DEGs) or differentially expressed miRNAs (DEMs) in a given disease [11, 12]. At the same time, leveraging integrated dual-disease analysis strategies based on publicly available datasets has facilitated the comprehension of shared molecular mechanisms underpinning disease pathophysiology, fostering the discovery of novel diagnostic markers and promising therapeutic targets. For instance, research findings have unveiled that a cluster of four genes (MMP12, PLAU, KRT14, and DKK1) may be involved in the common pathogenetic mechanism of craniopharyngioma and type 2 diabetes [13]. Furthermore, a recent investigation reported that viral gene replication and response to type I interferon may be the crucial bridge between dermatomyositis and nasopharyngeal carcinoma [14]. Likewise, a study exploring common gene signatures in pancreatic cancer and type 2 diabetes identified the regulation of endodermal cell fate specification as a plausible shared pathogenic mechanism, pinpointing S100A6 as a promising biomarker and therapeutic target for patients affected by both diseases [15].

In the present study, we aimed to identify the pivotal shared genes and miRNAs between AP and EBV infection and to construct a miRNA-mRNA network by integrating two mRNA and two miRNA datasets. Furthermore, we delved into the biological functions of these biomarkers, their relationship with immune cell infiltration, their diagnostic performance, and performed drug prediction analysis. We also constructed a competitive endogenous RNA (ceRNA) network and a transcription factor (TF)-miRNA-mRNA network to unveil the intricate upstream and downstream regulatory interconnections.

## Materials and methods

### Data collection

The mRNA and miRNA expression profiling data and related grouping information were gathered from Gene Expression Omnibus (GEO) (http://www.ncbi.nlm.nih.gov/geo/) database utilizing the search terms "acute pancreatitis", "Epstein-Barr virus infection", "EBV infection", and "infectious mononucleosis". The criteria for dataset selection were as follows: the dataset must contain samples from both the disease group and the control group; the tissue used for sequencing in different datasets should be as consistent as possible; if there are several homogeneous datasets, the dataset with the largest sample size is preferred. After careful screening, two mRNA datasets GSE194331 [16] and GSE45918 [17], alongside two miRNA datasets GSE42455 [18] and GSE109220 [19] with their corresponding platform annotation

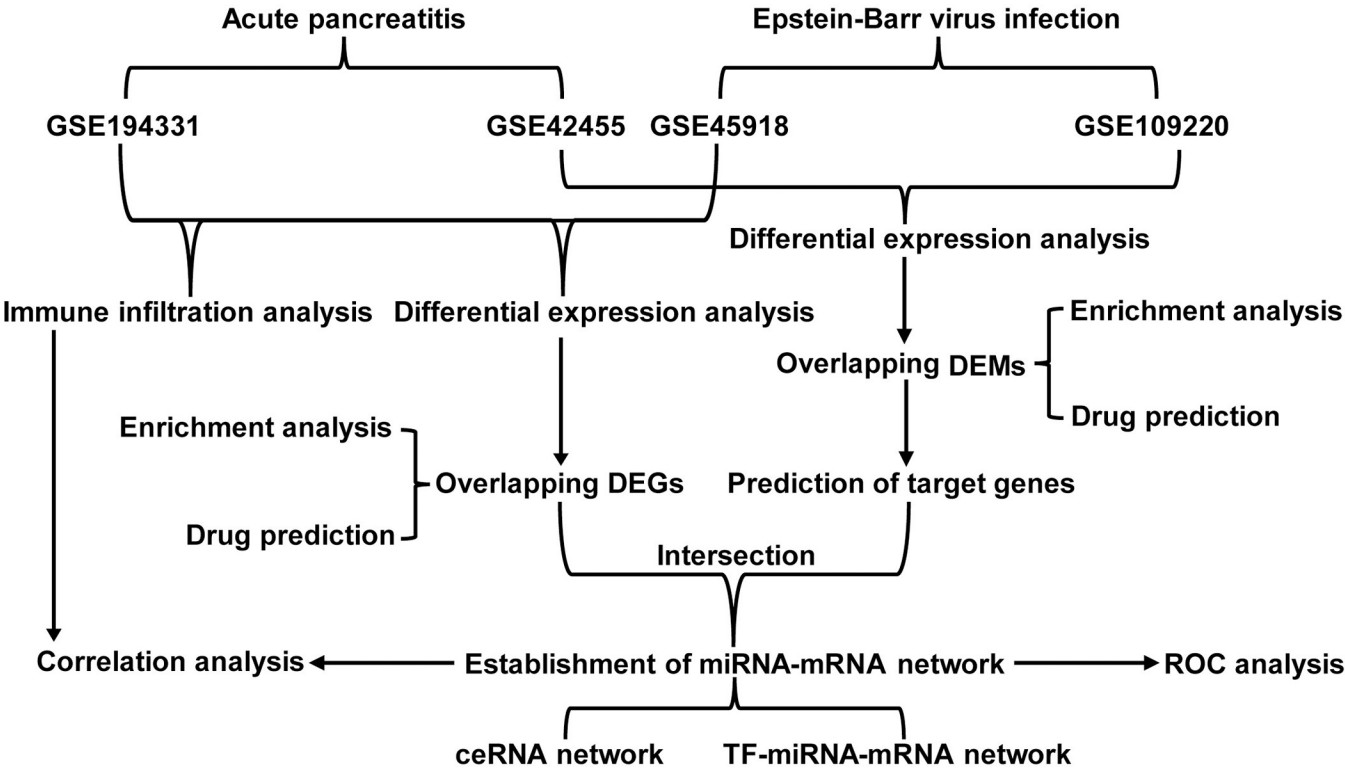

**Fig 1. Flowchart of the data analysis procedure.**

files were selected and downloaded. Dataset GSE194331 performed RNA-Seq of peripheral blood collected from 87 patients with AP of varying severity (mild = 57, moderately-severe = 20, severe = 10) and from 32 healthy controls. Dataset GSE45918 performed expression profiling by array of human peripheral blood before and after acquisition of EBV infection. Dataset GSE42455 performed non-coding RNA profiling by array of mesenteric lymph samples from AP rat models, which encompassed three distinct cohorts: control sham operated, fluid resuscitated taurocholate induced acute pancreatitis, and non-resuscitated taurocholate induced pancreatitis groups. The latter group represented a more severe form of the disease. Dataset GSE109220 performed non-coding RNA profiling by array of peripheral blood samples from symptomatic infectious mononucleosis patients at diagnosis, 1, 2, and 7 months post-diagnosis, alongside samples from 3 healthy controls. This study is based on the public datasets in the GEO database, which does not contain the specific information of the subjects and their personal privacy. Therefore, we believe that this study does not require ethics approval. The workflow of this study was elucidated in Fig 1. Detailed information on these datasets was listed in S1 Table.

## Data pre-processing

For dataset GSE194331, matrix of raw counts was downloaded and ensembl id was converted to gene symbol using the bitr() function of the clusterProfiler (version 4.12.0) package [20]. For dataset GSE45918, non-normalized matrix was downloaded and the neqc() function of the limma (version 3.60.2) package [21] was used for background correction and quantile normalization. Matrix files of datasets GSE42455 and GSE109220 were also downloaded. Probe id was converted to gene symbol or miRNA name with corresponding platform annotation files.

## Differential expression analysis

In dataset GSE194331, PANWES003, one of the mild AP patients, was identified as coming from a female patient who clustered with the males on the principal component analysis plot and was thus removed from any downstream analysis (see original article Methods). Additionally, the research findings highlighted that EBV-associated AP typically exhibited a favorable prognosis, with complications leaning towards the milder end of the spectrum [3, 4]. Hence, in alignment with this insight, 10 severe acute pancreatitis patients were also excluded from any further downstream analysis. Finally, the DESeq2 (version 1.44.0) package [22], which provides methods to test for differential expression by use of negative binomial generalized linear models on un-normalized RNA-seq counts data, was employed to identify the DEGs between 76 AP patients and 32 healthy controls in the dataset GSE194331. The limma is an R package for the analysis of gene expression data, especially the use of linear models for assessment of differential expression on normalized microarray data, which was used to identify the DEGs across 8 samples before and after acquisition of EBV infection in the dataset GSE45918. Similarly, in the dataset GSE109220, the limma package was used to identify the DEMs between 5 infectious mononucleosis diagnosis samples and 3 healthy controls. Within the dataset GSE42455, the limma package was employed for DEMs identification between 10 AP rat models and 5 sham samples.

The choice of the specific cutoff value will generally be based on the experimental design and the objectives of the study. The cutoff value of fold change is usually set to 1.3, 1.5 or 2 [16, 23, 24], with a corrected p-value < 0.05 [16], sometimes a p-value < 0.05 [25]. In order to adequately identify more DEGs and DEMs in the present study, we set the cutoff value of fold change to 1.5, meaning the differential gene or miRNA for which the expression level of the disease group samples was up- or down-regulated by a factor of 1.5 compared with that of the control samples, and it is also necessary to satisfy the corrected p-value < 0.05 in the datasets GSE194331, GSE45918 and GSE109220. Moreover, a slight adjustment was made to the threshold criteria of significance level, setting it at a p-value < 0.05 within the dataset GSE42455. This was because only 11 DEMs were identified using the corrected p-value < 0.05 as the threshold. The volcano plots of DEGs and DEMs were generated with ggplot2 (version 3.5.1) package.

## Enrichment analysis of DEGs and DEMs

Enrichment analysis helps researchers gain mechanistic insight into gene or miRNA lists generated from genome-scale experiments. The overlapping DEGs were subjected to Gene Ontology (GO), Kyoto Encyclopedia of Genes and Genomes (KEGG) pathway and disease ontology enrichment analysis using the clusterProfiler and DOSE (version 3.30.1) [26] packages at the threshold for adjusted p-value < 0.05. The overlapping DEMs were subjected to enrichment analysis using the miRNA Enrichment Analysis and Annotation Tool (version 2.1) [27] (miEAA, https://ccb-compute2.cs.uni-saarland.de/mieaa/) with default parameters. The enrichment results were downloaded and then visualized using the ggplot2 package.

## Establishment of miRNA-mRNA network

The get_multimir() function of the multiMiR (version 1.26.0) package [28] was employed to retrieve the DEMs target genes. The multiMiR contains a wide collection of validated and predicted miRNA–target interactions and their associations with drugs and diseases. It is composed of 14 external databases, which include 3 validated, 8 predicted, and 3 drug- or disease-related miRNA-target databases. In order to more accurately predict target genes of a given miRNA, we only considered validated miRNA-target interactions from three databases (miRecords, miRTarBase and TarBase). According to the detailed information provided by the

multiMiR website (http://multimir.org/), there are 3045, 595913 and 644186 miRNA-target interactions records from miRecords, miRTarBase and TarBase, respectively. Given that the miRecords database contains approximately 3000 records, significantly fewer than miRTar-Base and TarBase, and has not been updated since 2013, we opted to focus on miRNA-target gene interactions present in a minimum of two databases instead of requiring overlap across all three databases to ascertain the ultimate target genes. Subsequently, the predicted target genes were intersected with overlapping DEMs from the datasets GSE194331 and GSE45918 to identify candidate genes. Then a miRNA-mRNA regulatory network was constructed using the ggnetwork (version 0.5.13) and network (version 1.18.2) packages.

## Construction of the ceRNA network

Based on the miRNA-mRNA interactions, we sought to construct a ceRNA network. The interactions of miRNA with possible lncRNA and circRNA were identified by using miRNA-Target module with Encyclopedia of RNA Interactomes (ENCORI/starBase, https://rnasysu.com/encori/index.php) [29]. ENCORI/starBase is an openly licensed and state-of-the-art platform to facilitate the integrative, interactive and versatile display of, as well as the comprehensive annotation and discovery of, RNA-RNA and protein-RNA interactions by deeply mining thousands of high-throughput sequencing data of RNA-RNA interactome, CLIP-seq and degradome sequencing. By inputting specific miRNA, miRNA-mRNA, miRNA-lncRNA, and miRNA-circRNA interactions predictions can be performed step by step. In order to simplify the interaction links for visualization, the interactions results were sorted by the ClipExpNum, which represented the number of supported experiments, with low number indicating weak interaction with the miRNA [25]. Therefore, the corresponding top 10 lncRNAs and circRNAs candidates were selected, and the lncRNA-miRNA-mRNA and circRNA-miRNA-mRNA networks were constructed with ggnetwork and network packages.

## Construction of the transcription factor (TF)-miRNA-mRNA network

TF binds to the DNA and regulates transcription, affecting the increase or decrease of miRNA or mRNA levels. Firstly, TF-miRNA regulations were identified by using TransmiR v2.0 database (http://www.cuilab.cn/transmir) [30]. Only TF-miRNA regulations based on level 2 evidence or literature curation were included. Secondly, TF-mRNA regulations were identified by using hTFtarget database (https://guolab.wchscu.cn/hTFtarget/) [31]. Finally, overlapping TFs from TF-miRNA and TF-mRNA regulations were screened for TF-miRNA-mRNA network construction. Regulations between TFs and miRNAs/mRNAs were visualized with ComplexHeatmap (version 2.20.0) package [32]. The TF-miRNA-mRNA network was constructed using Cytoscape software (version 3.9.1) (https://cytoscape.org/).

## Landscape of immune cell infiltration

The single-sample Gene Set Enrichment Analysis (ssGSEA) from GSVA (version 1.52.3) package [33] was employed to investigate the abundance of 28 immune cells between patients and control samples in the datasets GSE194331 and GSE45918. Corresponding gene markers of immune cells were derived from previous literature [34]. The correlation of DEGs with the content of the immune cells was calculated using the Pearson method.

## Drug prediction

The Connectivity Map (CMap, https://www.broadinstitute.org/connectivity-map-cmap) [35] integrates large perturbational datasets to advance our understanding of disease and accelerate

the discovery of novel therapeutics. The query tool of cloud-based software platform termed CMap and LINCS Unified Environment (CLUE, https://clue.io/query) was used for drug prediction, mode of action analysis (MOA), and drug-target analysis. Briefly, the query gene set were compared with the reference data from the database to obtain a connectivity score (-100 to 100). A positive score indicates there is similarity between a given perturbagen's signature and that of the query, while a negative score indicates that the two signatures are opposing. Overlapping DEGs were subjected to query tool with specific parameters configured as "Gene expression (L1000)", "Touchstone", "Individual query", and "1.0". The results were downloaded and sorted by connectivity score, and top 20 small molecule compounds with the most negative connectivity scores were visualized by ggsankey (version 0.0.99999) and ggplot2 packages.

The dysregulated expression of miRNAs plays a pivotal role in disease pathogenesis, underscoring the potential of miRNA-targeted therapies as a novel approach for treating human diseases. The miRNA-drug associations were identified by using the multiMiR package. The miRNA-drug network was also constructed and visualized with ggnetwork package.

## Statistical analysis

All statistical analysis was performed using R software (version 4.4.0). The t-test was used to compare the data between two groups, and Pearson correlation analysis was employed to determine the correlation between the variables. The diagnostic efficacy was calculated by receiver operator characteristic (ROC) curves and area under the curve (AUC) using the pROC (version 1.18.5) package in R. Unless specifically mentioned, p-value < 0.05 was considered statistically significant.

## Results

### Identification of DEGs and enrichment analysis

In the dataset GSE194331, a sum of 3434 DEGs were identified through contrasting the AP group and the control group, of which 2270 were up-regulated and 1164 were down-regulated in the AP group (Fig 2A). Within the dataset GSE45918, a total of 1494 DEGs were screened by comparing the samples before and after acquisition of EBV infection, of which 673 were up-regulated and 821 were down-regulated after EBV infection (Fig 2B). Upon computing the intersection of DEGs from the datasets GSE194331 and GSE45918, a cumulative of 111 DEGs, comprising 42 up-regulated and 69 down-regulated genes, were recognized as overlapping DEGs (Fig 2C and S2 Table).

Then, we performed the GO, KEGG and disease ontology enrichment analysis. Utilizing the 42 up-regulated DEGs as inputs, a collective of 34 biological process (BP) and 7 KEGG terms were significantly enriched (Fig 2D and S3 Table), mainly comprising innate and adaptive immune response, interferon-gamma, complement and apoptotic signaling pathways. Using the 69 down-regulated DEGs as input, a total of 4 BP, 2 KEGG and 15 disease ontology terms were significantly enriched (Fig 2E and S4 Table), which mainly included B cell proliferation, immune response, autoimmune diseases and tumors.

### Identification of DEMs and enrichment analysis

In the dataset GSE42455, a total of 46 DEMs were identified by comparing the AP group and the control sham group, of which 17 were up-regulated and 29 were down-regulated in the AP group (Fig 3A). In the dataset GSE109220, a sum of 321 DEMs were screened by comparing the diagnosis group and healthy controls, of which 259 were up-regulated and 62 were down-

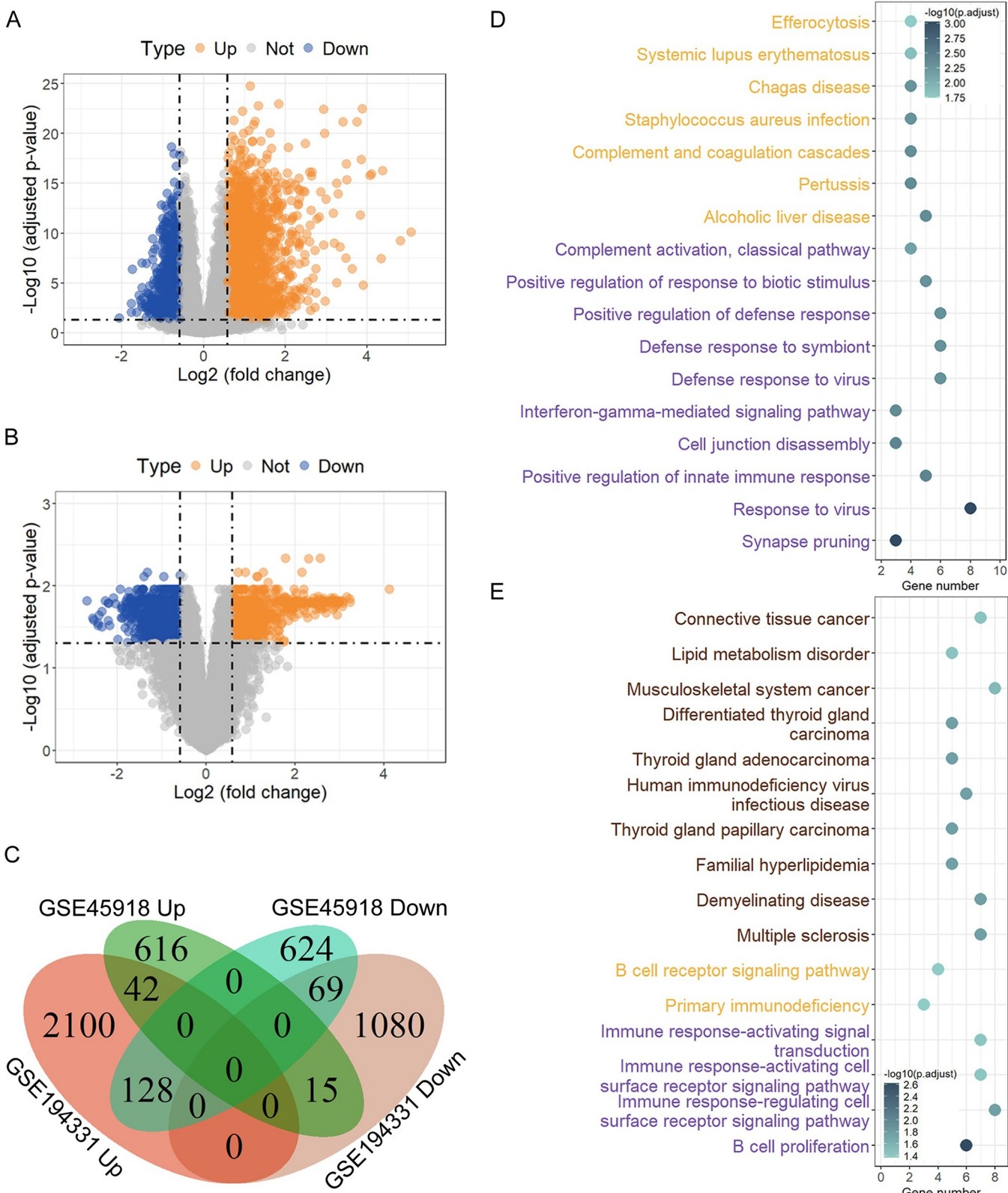

**Fig 2. Identification of DEGs and enrichment analysis.** The Volcano plots of DEG distributions from datasets GSE194331 (A) and GSE45918 (B). Up-regulated genes are marked by orange points, down-regulated genes are marked by blue points, and genes with no significant differences are marked by grey

points. (C) Venn diagram of shared DEGs between datasets GSE194331 and GSE45918. Top 10 enriched GO, KEGG, and disease ontology terms (if available) of the shared up-regulated genes (D) and down-regulated genes (E) from C. GO, KEGG, and disease ontology terms are marked in dark orchid, orange, and saddle brown, respectively.

regulated in EBV diagnosis group (Fig 3B). Upon computing the intersection of DEMs from the datasets GSE42455 and GSE109220, a total of 8 DEMs, including 1 up-regulated (miR-363-3p) and 7 down-regulated (miR-27a-5p, miR-130a-3p, miR-99b-5p, miR-23b-3p, miR-324-5p, miR-24-3p, miR-497-5p) miRNAs, were identified as overlapping DEMs (Fig 3C).

Using the miEAA tool, a total of 234 subcategories were significantly enriched based on 8 DEMs (Fig 3D and S5 Table). GO and KEGG subcategories mainly include cell cycle, nutrient metabolism and organ development. The diseases subcategories mainly encompass a wide range of solid and hematological tumors, neurological disorders and autoimmune diseases (Fig 3E and S5 Table).

## Establishment of miRNA-mRNA network

Using the 8 overlapping DEMs as input, a total of 25, 1057, and 1055 validated miRNA-target interactions were found in miRecords, miRTarBase, and TarBase databases by multiMiR package, respectively (S6 Table). Then 960 target genes identified in at least two databases were further selected (Fig 4A). A total of 10 overlapping genes were identified by taking intersections in 960 target genes and 111 DEGs (Fig 4B). Finally, a miRNA-mRNA network of 11 miRNA–mRNA interaction pairs was constructed, which included 5 DEMs: miR-24-3p, miR-23b-3p, miR-130a-3p, miR-324-5p, miR-497-5p, and 10 DEGs: E2F2, HMGB2, LDHA, UBE2C, LMNB1, CEP55, AHNAK, AKT3, PLAG1, PLEKHA1 (Fig 4C).

## CeRNA network construction

Non-coding RNAs, encompassing circRNA, lncRNA, and miRNA, are pivotal regulators of gene expression at the transcriptional and post-transcriptional level. For a more profound comprehension of the regulatory framework concerning the 11 miRNA–mRNA pairs unveiled in the above analysis, we constructed a ceRNA network. The results about lncRNA-miRNA-mRNA network and circRNA-miRNA-mRNA network were shown in Fig 5 (S7 Table). These findings afford an initial glimpse into the regulatory network of our identified hub miRNAs.

## TF-miRNA-mRNA network construction

TF is a master transcriptional regulator that affects the level of miRNA or mRNA. Using the 5 overlapping DEMs as input, a total of 140 TF-miRNA regulations with 107 TFs were found by the TransmiR v2.0 database. Using the 10 overlapping DEGs as input, a sum of 676 TF-mRNA regulations with 195 TFs were found by the hTFtarget database. Sixty-four overlapping TFs were identified by taking intersections. Among them, E2F1, MAZ, BRD4, CDK9, CEBPB, CREB1, ETS1, HDAC2, JUN, KLF5, and EP300 regulated almost all 10 overlapping DEGs, E2F1, MYC, and HIF1A regulated almost all 5 overlapping DEMs (Fig 6A). Finally, a TF-miRNA-mRNA network comprising 64 TFs, 5 miRNAs, and 10 genes was constructed by using Cytoscape (Fig 6B).

## Immune cell infiltration and its relationship with identified genes

The significantly enriched immune response made it necessary to investigate the possible dys-regulation of immune cells in AP and EBV infection. The results from ssGSEA analysis showed that there were significant abundance differences in multiple immune cell populations

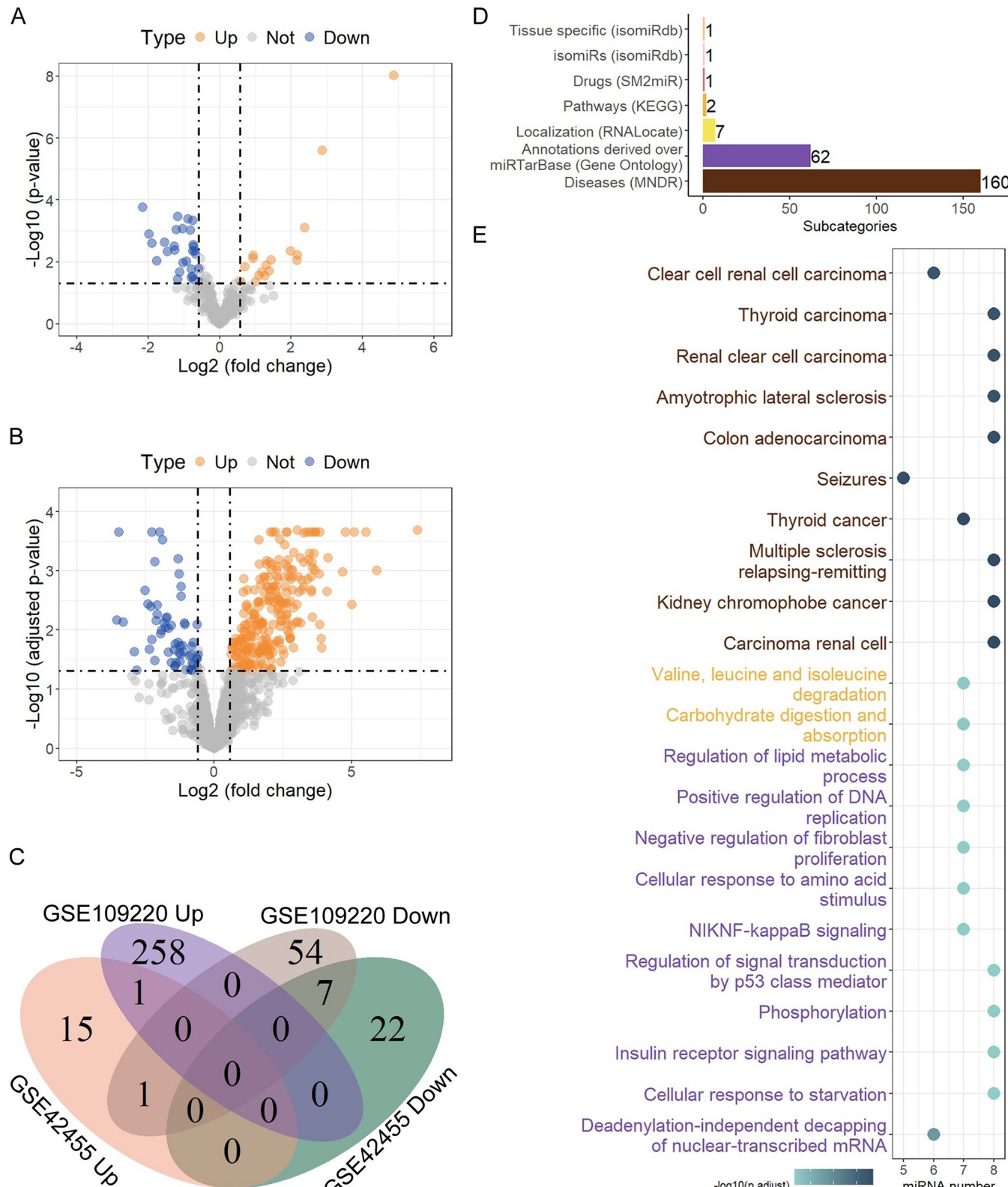

**Fig 3. Identification of DEMs and enrichment analysis.** The Volcano plots of DEM distributions from datasets GSE42455 (A) and GSE109220 (B). Up-regulated miRNAs are marked by orange points, down-regulated miRNAs are marked by blue points, and miRNAs with no significant differences are marked by grey points. (C) Venn diagram of shared DEMs between datasets GSE42455 and GSE109220. (D) Bar plot of enriched categories distributions of shared

DEMs from C. Numbers represent the corresponding subcategories. (E) Top 10 enriched GO, KEGG, and disease subcategories (if available) of the shared DEMs. GO, KEGG, and disease ontology subcategories are marked in dark orchid, orange, and saddle brown, respectively.

between diseased samples and controls (Fig 7A). Notably, a higher abundance of activated B cell was found in the control group in both datasets.

Moreover, we delved into the correlation between the 10 DEGs within miRNA-mRNA network and significantly different immune cell populations in both datasets. The results basically showed that the six up-regulated genes (CEP55, E2F2, HMGB2, LDHA, LMNB1, and UBE2C) were positively correlated with immune cells that exhibited higher abundance in AP or EBV infection patients compared to controls, and negatively correlated with immune cells that exhibited lower abundance in AP or EBV infection patients than in controls (Fig 7B and 7C). The converse held true for the four down-regulated genes (AHNAK, AKT3, PLAG1, and PLEKHA1) (Fig 7B and 7C).

## Diagnostic performance of identified biomarkers

Due to the clinical accessibility of blood samples, we conducted ROC and AUC analysis to evaluate the diagnostic performance of the identified biomarkers in three peripheral blood-derived datasets. In the dataset GSE194331, the AUC values of 10 hub genes exceeded 0.82, with exceptions noted for CEP55, PLAG1, and E2F2 (Fig 8A). Within the dataset GSE45918, all 10 hub genes displayed AUC values surpassing 0.87, notably showcasing the highest AUC with the genes PLEKHA1 and LMNB1, both achieving a perfect score of 1 (Fig 8B). In the dataset GSE109220, the AUC values of all 5 identified miRNAs reached 1, demonstrating a robust diagnostic performance (Fig 8C).

## Drug prediction

The management of diseases has perennially stood out as a focal point in clinical settings, propelling us to investigate promising small molecule compounds or drugs predicated on these perturbed genes and miRNAs. Through CMap analysis, a pool of 2428 compounds with their corresponding connectivity scores was generated (S8 Table). The top 20 compounds exhibiting the most negative connectivity scores along with their targets and MOA were depicted in Fig 9A. Notably, among them, dacinostat, panobinostat, vorinostat, pyroxamide, THM-I-94, entinostat, scriptaid, apicidin, and ISOX were pinpointed as histone deacetylase (HDAC) inhibitors. The results of miRNA-drug associations from the multiMiR unveiled a total of 7 drugs identified as prospective therapeutic remedies (Fig 9B).

## Discussion

An increasing number of studies have shown a robust correlation between EBV infection and the onset of AP [3–5, 10]. Nevertheless, the intricate molecular link is not clear. In present study, for the first time, we uncovered a shared miRNA-mRNA network between AP and EBV infection, providing a reference for further research.

The majority of studies to date examining the impact of EBV infection have focused on the mRNA or miRNA profile within the virally infected cell and the implications for tumorigenesis. In contrast, we analyzed the impact of EBV infection on the mRNA and miRNA profiles in peripheral lymphoid cells that typically participate in the immune response to EBV and integrated relevant datasets derived from peripheral blood cells and mesenteric lymph in the context of acute pancreatitis to investigate molecular biological mechanisms shared in the both disease states. During the datasets screening process, we also found another dataset GSE109227 related with AP in the GEO website. In the dataset GSE109227 [36], RNA from

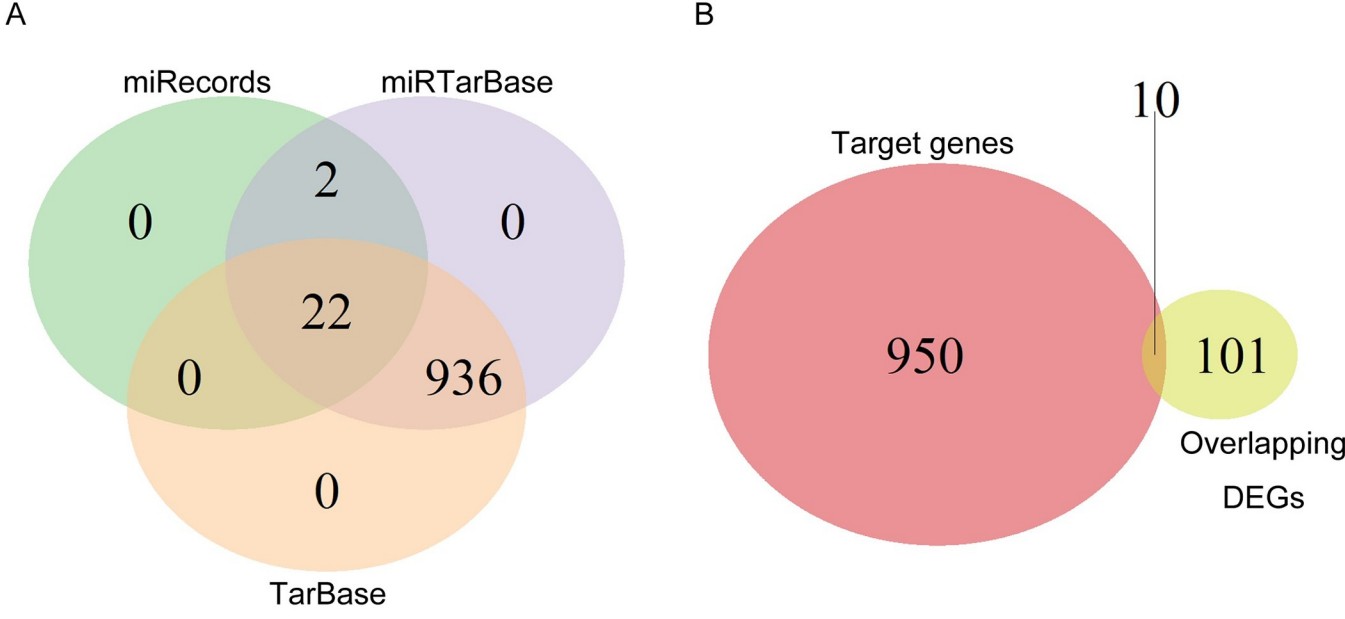

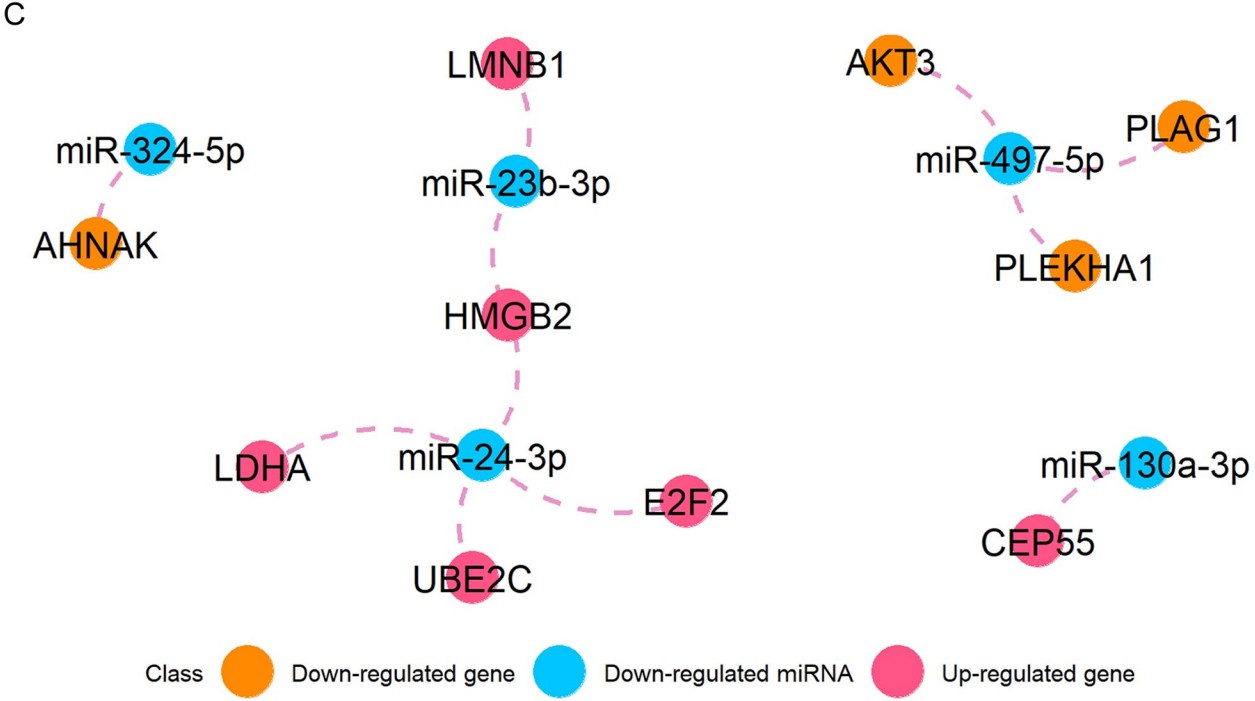

**Fig 4. Construction of miRNA-mRNA regulatory network.** (A) Venn diagram of overlapping target genes between miRecords, miRTarBase, and TarBase databases. (B) Venn diagram of shared genes between target genes and overlapping DEGs from Fig 2C. (C) A miRNA-mRNA regulatory network.

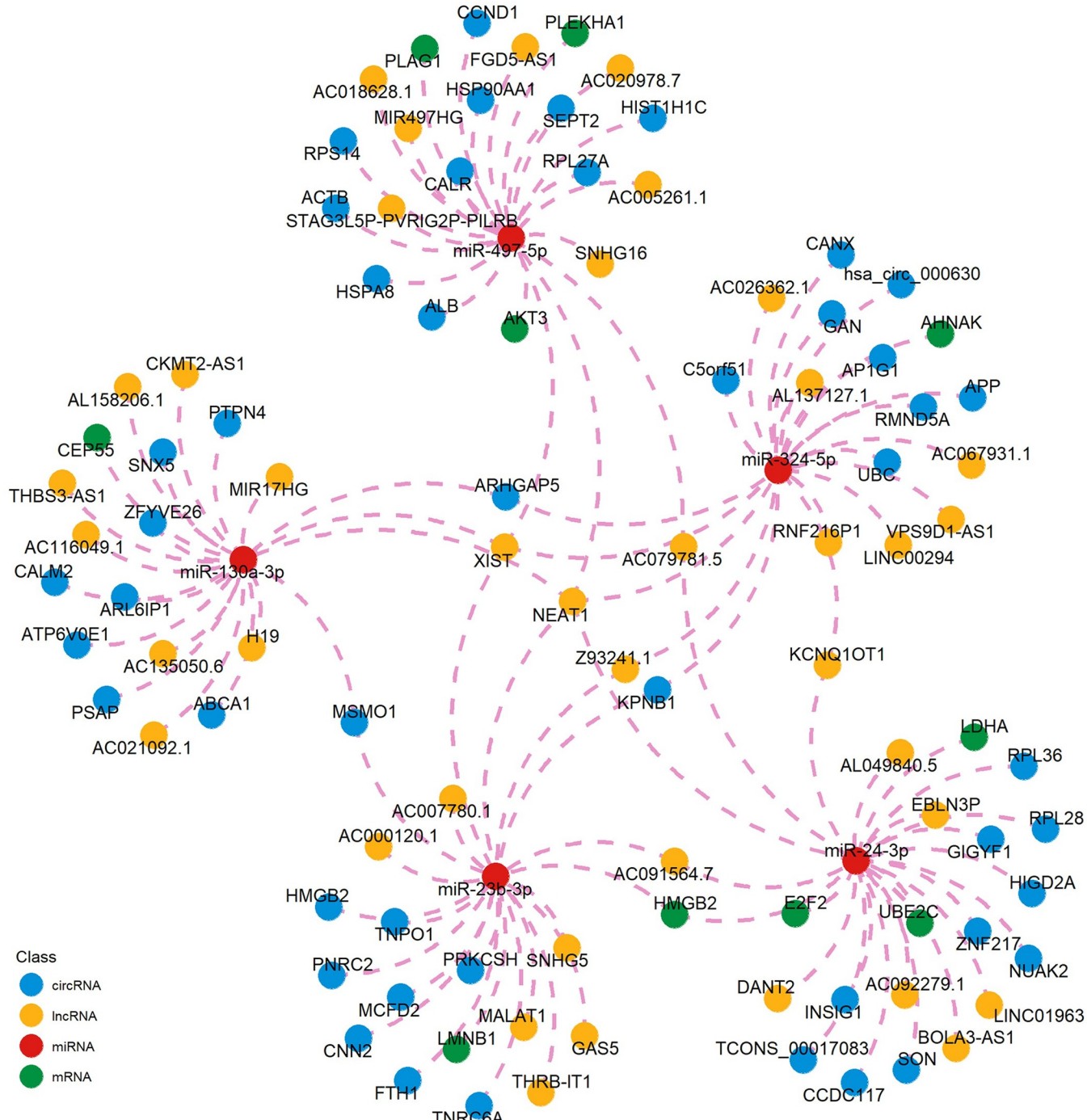

**Fig 5. Construction of the ceRNA network.** The network included 5 miRNAs, 10 genes, top 10 lncRNAs and circRNAs candidates of corresponding miRNAs.

cerulein induced AP mice pancreas was extracted and processed for microarray analysis and the authors used these tissues for screening new potential biomarkers of AP. Finally, they identified RCAN1 as a potential diagnostic marker of AP. Since there is no expression profile data of pancreas with EBV infection, and in order to make the tissues used for sequencing in different datasets as consistent as possible, the dataset GSE109227 was omitted.

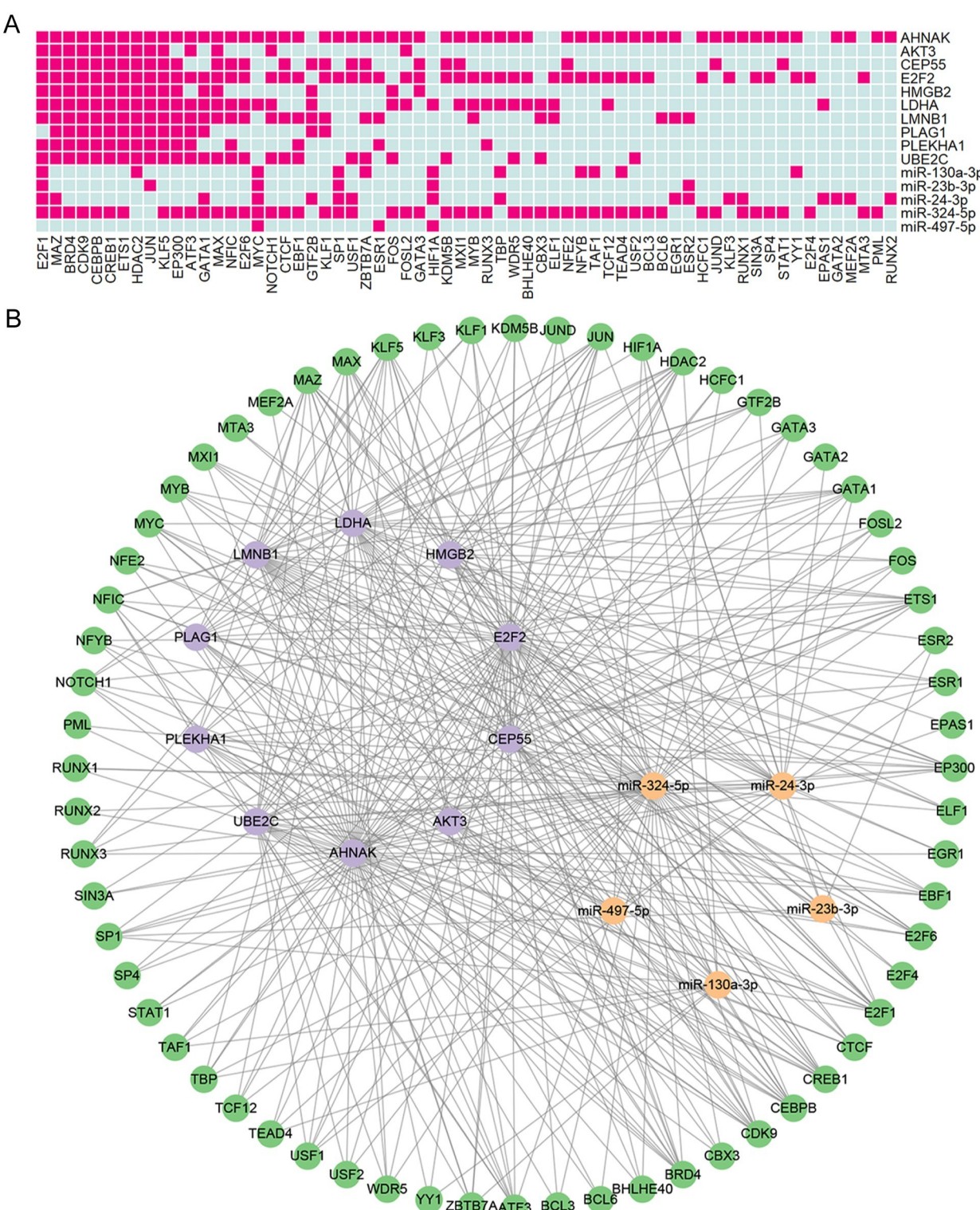

**Fig 6. Construction of the TF-miRNA-mRNA network.** (A) Heatmap of TF-miRNA regulations and TF-mRNA regulations. The dark pink cell represents the TF-miRNA/mRNA regulatory relationship. (B) A TF-miRNA-mRNA regulatory network. Green, purple and orange circles represent transcription factors, mRNAs, and miRNAs, respectively.

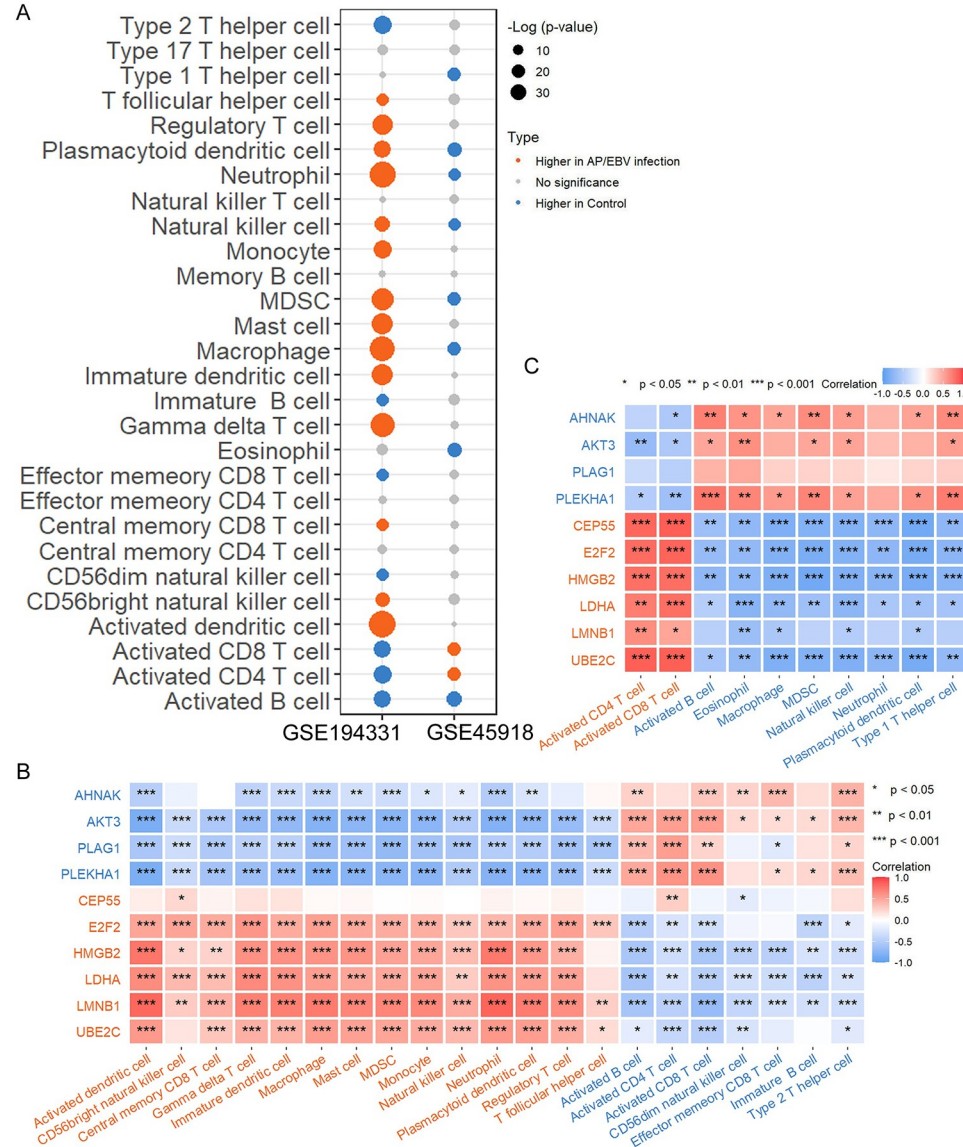

**Fig 7. Landscape of immune cell infiltration.** (A) Bubble plot of estimated abundance of 28 immune cells between patients and control samples in datasets GSE194331 and GSE45918. Higher abundance of immune cells in AP or after EBV infection samples are marked by coral dots, higher abundance of immune cells in control samples are marked by royal blue dots, abundance of immune cells with no significant differences are marked by grey dots. Correlation heatmap of 10 DEGs and significant difference immune cells in GSE194331 (B) and GSE45918 (C). Positive and negative correlation was respectively shown in coral and royal blue color. Up-regulated genes and higher abundance of immune cells in AP or after EBV infection samples are marked by coral, and down-regulated genes and higher abundance of immune cells in control samples are marked by royal blue. * $p < 0.05$, ** $p < 0.01$, *** $p < 0.001$.

Functional enrichment analysis showed that shared DEGs and DEMs between AP and EBV infection were involved in various biological processes and signaling pathways, such as innate and adaptive immune responses, autophagy and apoptosis signaling pathways. Innate immunity is the first line of antiviral defense and the activation of interferon signaling pathway is one of the major antiviral defense mechanisms of host cells after EBV infection [17, 37]. Consistent with our enrichment analysis results, a recent study also revealed that AP mice were enriched in biological processes related to immune system pathways and inflammatory

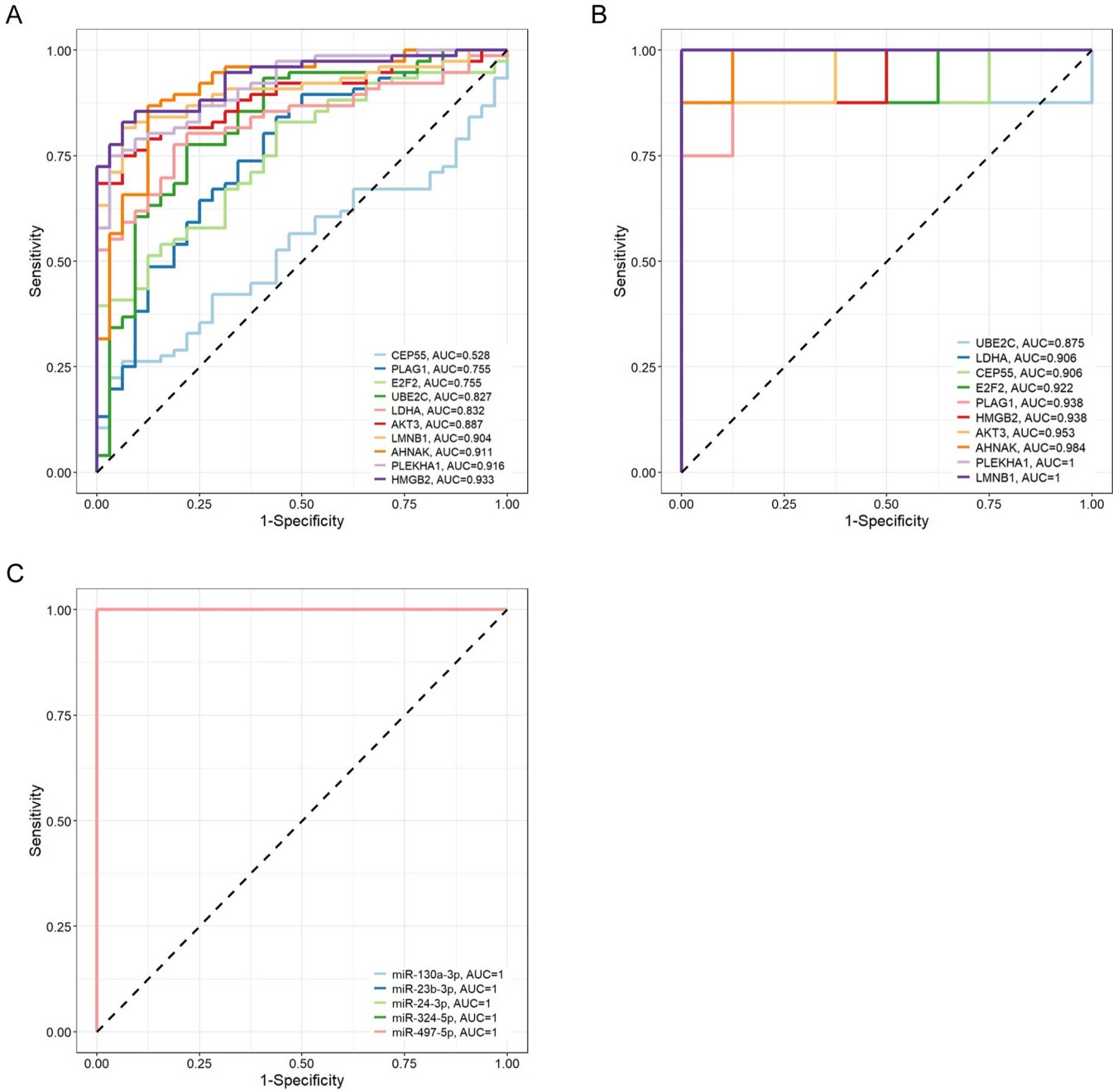

**Fig 8. Evaluation of the diagnostic performance of identified biomarkers.** ROC curve analysis hub genes in the datasets GSE194331 (A), GSE45918 (B), and GSE109220(C).

responses [38]. However, innate immunity is also a double-edged sword, as the induction of pro-inflammatory responses and the activation of programmed cell death [37]. The patho-physiology of EBV-associated pancreatitis is unclear, but the pancreas is likely one of the target organs affected by the immune response. The unobstructed autophagy process is the basis for cells to maintain their metabolic functions, and the abnormal autophagy can cause autophagic vesicles to accumulation and lysosome inactivation, which is closely related to the occurrence

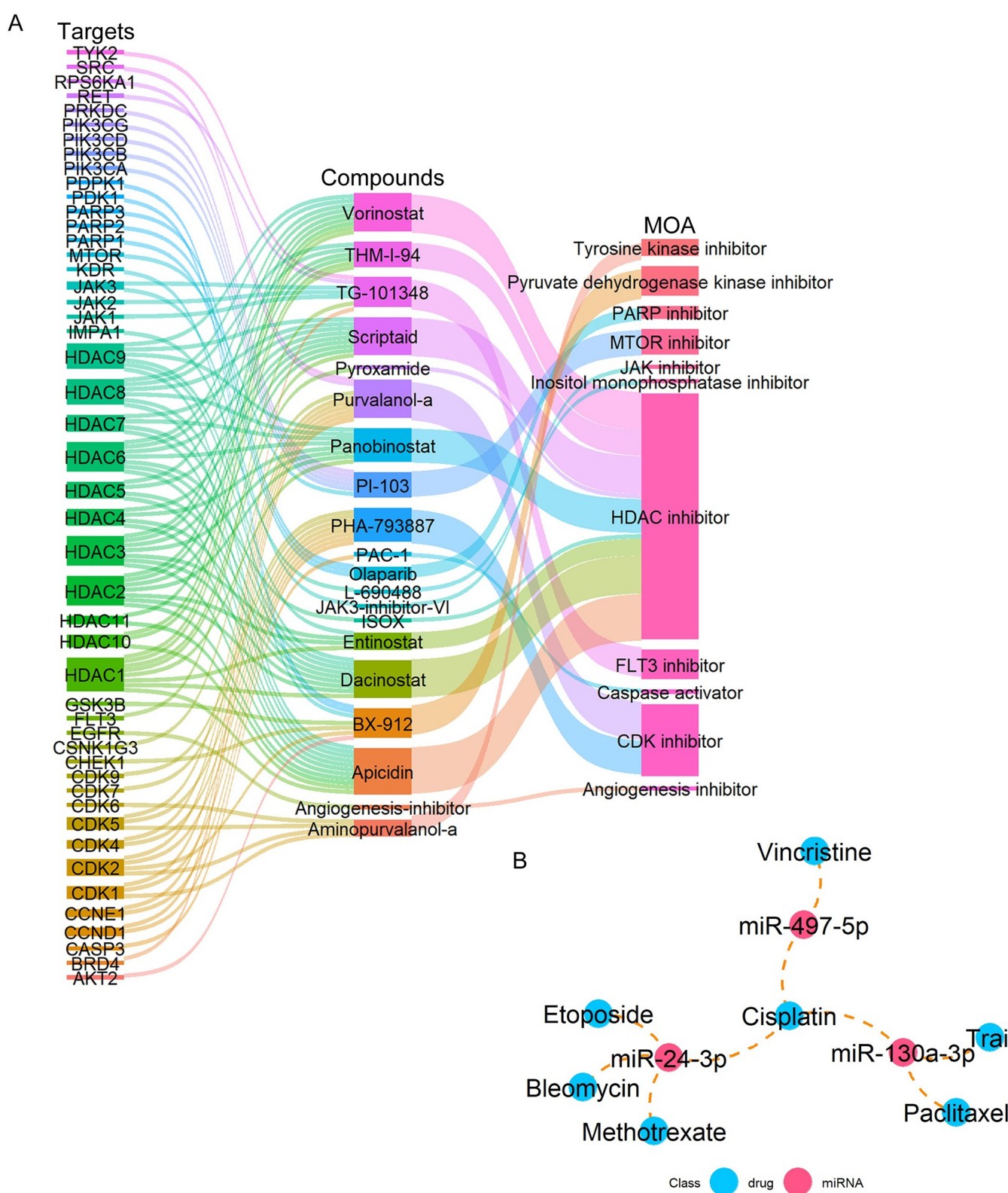

**Fig 9. Drug prediction based on the shared DEGs and DEMs.** (A) Sankey diagram of top 20 compounds (middle column) with corresponding target genes (left column) and MOA (right column). (B) A drug-miRNA network. The drug-miRNA associations were integrated from the multiMiR package.

of pancreatitis [39]. Studies also find that EBV have evolved multiple strategies to interfere with autophagy to avoid destruction and promote their own replication and spread [40, 41]. Increasing evidences indicate that EBV miRNAs and latent genes involved in manipulation of the cell apoptosis [42, 43]. Mild acute pancreatitis was found to be associated with extensive apoptotic acinar cell death, whereas severe AP was found to involve extensive acinar cell necrosis but very little acinar cell apoptosis [44]. Consistent with this, the results of our enrichment analysis found that apoptosis, but not necrosis, was significantly enriched. And the patients with EBV-associated AP were characterized by a favorable prognosis, without severe complications. However, since the datasets used for DEG and DEM identification are not from pancreatic tissue, it is still unclear which exact molecules and pathways are affected in the pancreas during EBV infection, and whether EBV can directly cause pancreatic cell damage and its potential molecular mechanisms still require further co-culture and in vivo studies.

In order to showcase the upstream and downstream regulatory relationships of hub molecules, the ceRNA and TF-miRNA-mRNA networks were constructed. We found that lncRNA NEAT1 exhibited close relationships with five identified hub miRNAs in the ceRNA network. There are no reported associations between NEAT1 and EBV infection, but studies have shown that NEAT1 has opposite effects in limiting the replication of dengue and Hantaan viruses [45, 46]. Downregulation of NEAT1 relieves caerulein-induced cell apoptosis and inflammatory injury in pancreatic acinar cells through sponging miR-365a-3p [47]. The NEAT1/miR-130/IRF1 axis in gastric cancer [48], NEAT1/miR-24-3p/LRG1 axis in thoracic aortic aneurysm [49], NEAT1/miR-324-5p/RAN axis in retinoblastoma [50] and NEAT1/miR-497-5p/PIK3R1 axis in renal tubular oxidative injury [51] have been widely studied. E2F1 acts as a master regulator in the TF-miRNA-mRNA network, which is a member of the E2F family of transcription factors and plays a crucial role in the control of cell cycle and action of tumor suppressor proteins. Studies have found that EBV-encoded proteins induce activation of E2F1 and S-phase entry, which may facilitate efficient lytic EBV replication in certain cell types [52, 53]. A pan-cancer study indicated that KAT2A was found to cooperate with E2F1 and be recruited by E2F1 to the UBE2C promoter for elevating the expression of UBE2C by increasing the acetylation level of H3K9 to promote cell proliferation and the migration of cancer cells [54]. And increased E2F1 expression via the E2F1/FTH1P3/miR-377-3p/LDHA axis promoted cell viability and glycolysis in laryngeal squamous cell carcinoma [55]. In summary, although some studies have demonstrated the regulatory relationships in the ceRNA and TF-miRNA-mRNA networks, they still need to be further explored in the context of EBV infection and AP.

The results from immune cell infiltration analysis indicated that multiple immune cell populations were more abundant in AP samples than in controls, such as neutrophil and dendritic cells (DCs). Neutrophils are the first-line cells that migrate into the pancreatic inflammation site [56, 57], and are central to the development of pancreatitis, mediating local tissue damage in the pancreas as well as distant organ damage and subsequent death [58, 59]. Therapeutic interventions targeting neutrophils reduce tissue damage significantly and protect against the development of pancreatitis [59–61]. DCs are known to be essential for pancreatitis recovery [62]. Specifically, numbers of MHCII and CD11c dual-positive DCs increased 100-fold in pancreas of AP mice. Intrapancreatic DCs expressed higher levels of MHCII and CD86 and increased production of IL-6, membrane cofactor protein-1, and tumor necrosis factor-α. Depletion of DCs from mice with pancreatitis resulted in neutrophil infiltration and increased levels of systemic markers of inflammation, and died from acinar cell death within four days. A recent study distinguished six subsets of DCs in the pancreas and provided insights into their temporal behavior during pancreatitis [57]. DCs are a heterogeneous population and functional characterization of specific DCs subsets may advance our understanding of cell

states and responses in pancreatitis and recovery. Further work on this important issue is needed. In addition, we found a lower abundance of activated B cells in AP samples than in controls. The contribution of B cells to tissue injury and regeneration has been investigated in several experimental models [63–65]. Studies have shown that B cells suppress pancreatic regeneration, and their trafficking and/or function are modulated by hypoxia and HIF1α in the setting of pancreatitis [66]. Interestingly, HIF1A has also been identified in the TF-miRNA-mRNA network, which may provide new insights into the precise mechanism. EBV is commonly transmitted through infected saliva and infects resting B lymphocytes and epithelial cells [67]. EBV-infected naïve B lymphocytes undergo a germinal center-like activation and differentiation program [68]. Furthermore, antigen-presenting cells present antigens to T cells. The generation of EBV-specific CD8+ and CD4+ T cells is markedly elevated in individuals infected with EBV. Infected B cells are then attacked by cytotoxic T lymphocytes (CTLs), which control the number of infected B cells and play antiviral roles [67]. Prior research indicated a positive correlation between CD8+ lymphocytosis and disease severity [69]. Meanwhile, the eight patients in the dataset GSE45918 are all in serious condition (see original article Supporting Information S1 Table). Taken together, this is consistent with our immune infiltration analysis results, where we observed an increase in the abundance of activated CD8 and CD4 T cells and a decrease in activated B cells in the acute EBV-infected samples.

The basic concept of CMap is to use a reference database containing drug-specific gene expression profiles and compare it with a disease-specific gene signature, whose overall goal is to predict potentially therapeutic drug candidates [35]. HDAC inhibitors stand out in our Cmap drug prediction analysis, which target histone deacetylases, a family of enzymes responsible for removing acetyl groups from various protein substrates [70]. By inhibiting HDAC activity, these compounds promote histone acetylation, inducing alterations in chromatin configuration and gene transcription. HDAC inhibitors have emerged as promising therapeutic agents in the field of epigenetics, offering new avenues for the treatment of cancer, neurological disorders, viral infection, multiple sclerosis and other diseases [71]. Studies have shown that HDAC inhibition effectively reversed EBV-induced cancer cell dedifferentiation and suppressed tumor progression in nasopharyngeal carcinoma [72]. Furthermore, the HDAC inhibitor restored p53 apoptosis pathway, resulted in a necrosis/apoptosis switch and protected mice from experimentally induced AP [73]. HDAC inhibitors also promote pancreatic stellate cell apoptosis and relieve pancreatic fibrosis in chronic pancreatitis [74]. However, CMap also has certain constraints, such as limited drug perturbation data, a limited drug coverage, dosage-dependent conditions and the uncertainty of applying cell lines or animal model expression patterns to human systems [75]. For more comprehensive understanding the drug MoA, specificity, and off-target effects of HDAC inhibitors, some methodologies incorporating other omics than transcriptomics would be beneficial, including, for instance, methylation array, metabolomics and proteomics, as well as dynamic or longitudinal data, would widen the limited view captured by the single time point of transcriptomic responses, which will facilitate a shift in drug discovery towards a personalized and precision medicine treatment approach.

A total of 5 down-regulated miRNAs shared between AP and EBV infection were encapsulated in the miRNA-mRNA network, including miR-24-3p, miR-23b-3p, miR-130a-3p, miR-324-5p and miR-497-5p, with the AUC values reached 1 in the dataset GSE109220. A growing body of research suggests that miRNAs are strongly associated with a wide range of pathological condition or disease progression, such as inflammation, infection, and cancers [76–78]. Su et al. reported that an miR-24-3p/MARCH3/NLRP3 axis mediated peritoneal macrophage M1 polarization and pyroptosis in AP [79]. Studies suggested that miR-24-3p promotes porcine reproductive and respiratory syndrome virus replication through suppression of HO-1 expression [80]. The miR-23b-3p has a critical role in alleviating inflammation in Parkinson's disease

and epilepsy [81, 82]. Studies showed that miR-130a-3p/PPAR-γ axis participated in pancreatic stellate cell activation and collagen formation of chronic pancreatitis [83]. Ding et al. reported that miR-130a-3p could improve pulmonary fibrosis by downregulating the secretion of inflammatory cytokines and the deposition of extracellular matrix in the inflammatory and fibrotic phase, respectively [84]. It has been reported that miR-324-5p targets the viral PB1 gene and the host CUED2 gene to suppress H5N1 influenza A virus replication [85]. Subsequent studies demonstrate that interferon-mediated repression of miR-324-5p potentiates necroptosis to facilitate antiviral defense [86], suggesting that the mechanism of action of miR-324-5p in viral infection is context-dependent. Furthermore, miR-324-5p/BMPR2 axis was involved in the regulation of NF-κB signaling in osteoarthritis [87]. The dysregulation of miR-497-5p has been implicated in the pathogenesis of several diseases, with a strong emphasis on cancer. Studies have shown that miR-497-5p acts as a tumor suppressor in various cancer types by targeting oncogenes or genes involved in cell proliferation and survival [88, 89]. In addition, Zhang et al. found that miR-497-5p exerted an anti-inflammatory and anti-fibrotic effect in diabetic nephropathy [90]. Taken together, with the exception of miR-24-3p, there are no reports that the other four miRNAs are associated with AP or EBV infection; however, the above studies highlight the important role of these miRNAs in regulating inflammatory responses and provide a theoretical basis for further research in the context of AP or EBV infection.

A sum of 10 shared genes were encapsulated in the miRNA-mRNA network, most of which had AUC values greater than 0.8 in the datasets GSE194331 and GSE45918. Among them, three genes (HMGB2, LDHA and AKT3) were reported to be closely related to pancreatitis, and no gene was reported to be associated with EBV infection. The HMGB2 encodes a member of the non-histone chromosomal high mobility group protein family and plays a role in facilitating cooperative interactions between cis-acting proteins by promoting DNA flexibility. Su et al. reported that HMGB2 facilitated ACSL1 transcription, resulting in the activation of ferroptosis and thereby aggravating severe AP [91]. LDHA encodes the A subunit of lactate dehydrogenase enzyme, which is involved in lactate production and is a sensitive predictor of systemic complications and organ failure in AP [92, 93]. Li et al. demonstrated that metabolite lactate relieved macrophage-associated local and systemic inflammation of AP in a TLR4/MyD88- and NLRP3/caspase1-dependent manner [94]. The protein encoded by AKT is a member of the serine/threonine protein kinase family, which is involved in a wide variety of biological processes including cell proliferation, differentiation, apoptosis and tumorigenesis. Moreover, AKT3 regulated autophagy and was involved in inflammatory responses, apoptosis and angiogenesis in severe AP [95]. E2F2 is a member of the E2F family of transcription factors which plays an essential role in the control of cell cycle. UBE2C encodes a member of the E2 ubiquitin-conjugating enzyme family, which is required for the destruction of mitotic cyclins and for cell cycle progression. LMNB1 encodes one of the two B-type lamin proteins and is a component of the nuclear lamina. Gao et al. reported that LMNB1/CDKN1A signaling axis regulates the cell cycle in hepatocellular carcinoma [96]. CEP55 encodes a protein regulates spindle organization and cell cycle progression, its stabilization is required for normal execution of cytokinesis [97, 98]. Taken together, the above four genes are all up-regulated genes and are closely related to cell cycle regulation; cell cycle perturbation has been studied in the context of EBV infection [99] and AP [100], which may partly explain why they are significantly correlated with immune cells with differential abundance in the two groups. AHNAK is the giant jack of all trades, especially in calcium homeostasis [101]. Calcium overload is crucial in the pathogenesis of pancreatitis, which results in trypsin activation, vacuolization and necrosis [102]. In addition, $Ca^{2+}$ signaling plays an important role in B cell survival and activation following EBV infection [103]. The PLAG1 gene is located on chromosome 8q12 and is

known for its oncogenic properties [104]. Studies have found that Wnt signaling pathway is one of the downstream targets of PLAG1 [105]. Huang et al. reported that activation of the Wnt/β-catenin pathway could inhibit cell apoptosis and inflammatory cytokine release, thus improving pancreatic and intestinal damage in AP rats [106]. In addition, Wnt signaling pathway was involved in metastasis of EBV-associated gastric carcinoma [107]. PLEKHA1 encodes an adaptor protein that specifically binds phosphatidylinositol 3,4-bisphosphate and participates in lymphocyte activation [108, 109]. In summary, these genes are directly or indirectly involved in the molecular events in AP and EBV infection, providing new directions for further research.

However, the present study has several constraints. Initially, despite the amalgamation of four transcriptomic studies, certain datasets had limited sample sizes, potentially leading to biased results. Secondly, most transcriptomic data are derived from peripheral blood rather than pancreas, which may have certain limitations in explaining the onset of pancreatitis. Lastly, there is a deficiency of "wet" experiments for the validation of potential biomarkers. Consequently, additional clinical validations in a sizable patient cohort and experimental substantiations are imperative in the imminent future.

## Conclusions

To summarize, we firstly revealed a miRNA-mRNA network, comprising 5 miRNAs: miR-24-3p, miR-23b-3p, miR-130a-3p, miR-324-5p, miR-497-5p, and 10 genes: E2F2, HMGB2, LDHA, UBE2C, LMNB1, CEP55, AHNAK, AKT3, PLAG1, PLEKHA1, shared between AP and EBV infection, which may help to identify novel diagnostic biomarkers and therapeutic targets in clinical practice.

## Supporting information

**S1 Table. Detailed information about the platform and sample information of the included datasets (GSE194331, GSE45918, GSE42455 and GSE109220).**
(DOCX)

**S2 Table. 111 overlapping DEGs between the datasets GSE194331 and GSE45918.**
(DOCX)

**S3 Table. Enrichment analysis results based on the 42 overlapping up-regulated DEGs.**
(DOCX)

**S4 Table. Enrichment analysis results based on the 69 overlapping down-regulated DEGs.**
(DOCX)

**S5 Table. Enrichment analysis results based on the 8 overlapping DEMs from miEAA tool.**
(XLSX)

**S6 Table. Detailed information about the validated miRNA-target interactions in the miRecords, miRTarBase, and TarBase databases.**
(XLSX)

**S7 Table. Detailed information about the miRNA-circRNA and miRNA-lncRNA interactions in the ENCORI/starBase.**
(XLSX)

**S8 Table. Detailed information about the 2428 compounds from Cmap analysis.**
(XLSX)

## Acknowledgments

We thank the all authors of the datasets for uploading their data to public databases.

## Author Contributions

**Conceptualization:** Xing Wei.

**Data curation:** Xing Wei.

**Formal analysis:** Xing Wei.

**Investigation:** Xing Wei, Zhen Weng.

**Methodology:** Xing Wei, Zhen Weng.

**Software:** Xing Wei, Zhen Weng.

**Visualization:** Xing Wei.

**Writing – original draft:** Xing Wei, Xia Xu.

**Writing – review & editing:** Xia Xu, Jian Yao.

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
