## [Decision Letter · Decision Letter 0]

16 Aug 2024

PONE-D-24-30500Exploration of a miRNA-mRNA network shared between acute pancreatitis and Epstein-Barr virus infection by integrated bioinformatics analysisPLOS ONE

Dear Dr. Wei,

Thank you for submitting your manuscript to PLOS ONE. After careful consideration, we feel that it has merit but does not fully meet PLOS ONE’s publication criteria as it currently stands. Therefore, we invite you to submit a revised version of the manuscript that addresses the points raised during the review process.

 Please submit your revised manuscript by Sep 30 2024 11:59PM. If you will need more time than this to complete your revisions, please reply to this message or contact the journal office at plosone@plos.org. Please include the following items when submitting your revised manuscript:A rebuttal letter that responds to each point raised by the academic editor and reviewer(s). You should upload this letter as a separate file labeled 'Response to Reviewers'.A marked-up copy of your manuscript that highlights changes made to the original version. You should upload this as a separate file labeled 'Revised Manuscript with Track Changes'.An unmarked version of your revised paper without tracked changes. You should upload this as a separate file labeled 'Manuscript'.

We look forward to receiving your revised manuscript.

Kind regards,

Jinhui Liu

Academic Editor

PLOS ONE

Journal Requirements:

3. Please note that your Data Availability Statement is currently missing [the repository name and/or a direct link to access each database]. If your manuscript is accepted for publication, you will be asked to provide these details on a very short timeline. We therefore suggest that you provide this information now, though we will not hold up the peer review process if you are unable.

**Additional Editor Comments:**

Authors should revise according to the suggestions of reviewers. The modifications should be marked. A point to point response letter is needed.

Reviewers' comments:

Reviewer's Responses to Questions

**Comments to the Author**

1. Is the manuscript technically sound, and do the data support the conclusions?

Reviewer #1: Yes

Reviewer #2: Yes

Reviewer #3: No

2. Has the statistical analysis been performed appropriately and rigorously? 

Reviewer #1: Yes

Reviewer #2: Yes

Reviewer #3: No

3. Have the authors made all data underlying the findings in their manuscript fully available?

Reviewer #1: Yes

Reviewer #2: Yes

Reviewer #3: Yes

4. Is the manuscript presented in an intelligible fashion and written in standard English?

Reviewer #1: Yes

Reviewer #2: Yes

Reviewer #3: Yes

5. Review Comments to the Author

**Reviewer #1: **The manuscript titled "Exploration of a miRNA-mRNA network shared between acute pancreatitis and Epstein-Barr virus infection by integrated bioinformatics analysis" presents an interesting study that delves into the shared molecular mechanisms between acute pancreatitis (AP) and Epstein-Barr virus (EBV) infection. The use of integrated bioinformatics analysis to identify miRNA-mRNA networks is commendable, and the study's findings could contribute to a deeper understanding of these pathological conditions. However, several areas need improvement to enhance the manuscript's clarity, rigor, and overall impact.

Firstly, the manuscript would benefit from a more detailed explanation of the methods used in the differential expression analysis and network construction. While the authors have provided a general overview, the specific criteria for selecting differentially expressed genes (DEGs) and miRNAs (DEMs) and the rationale behind these choices need to be more clearly articulated. Additionally, the process of intersecting DEGs and DEMs to construct the miRNA-mRNA network should be described with greater precision, including any assumptions made or thresholds applied.

The enrichment analysis results, while comprehensive, could be more effectively integrated into the narrative. Currently, the findings from the Gene Ontology (GO), Kyoto Encyclopedia of Genes and Genomes (KEGG), and disease ontology analyses are presented in a somewhat fragmented manner. It would be beneficial to discuss these results in the context of the broader biological implications and how they relate to the known pathophysiology of AP and EBV infection. Furthermore, the study would benefit from a more critical discussion of the limitations of enrichment analysis, particularly the potential for over-interpretation of results due to the inherent biases in the databases used.

The construction of the ceRNA and TF-miRNA-mRNA networks is an important aspect of the study. However, the manuscript would be strengthened by a more detailed discussion of the biological relevance of the identified networks. Specifically, the authors should provide more insight into the potential functional roles of the miRNAs and genes within these networks, as well as any known associations with AP and EBV infection. Additionally, the criteria for selecting the top 10 lncRNAs and circRNAs in the ceRNA network should be clarified, as this selection appears somewhat arbitrary.

The immune cell infiltration analysis is a valuable addition to the study, but the interpretation of these results requires more depth. The manuscript should explore the implications of the observed differences in immune cell populations between AP and EBV-infected individuals and controls, and how these findings align with existing knowledge of immune responses in these conditions. Additionally, the correlation analysis between DEGs and immune cell populations could be expanded to include a discussion of potential mechanisms underlying these associations.

Regarding the drug prediction analysis, while the identification of histone deacetylase (HDAC) inhibitors as potential therapeutic agents is intriguing, the manuscript would benefit from a more nuanced discussion of the limitations of this approach. The authors should consider the potential challenges in translating these findings into clinical practice, including issues related to drug specificity, off-target effects, and the need for validation in experimental models.

**Reviewer #2: **As a reviewer, I have carefully examined the manuscript titled "Exploration of a miRNA-mRNA network shared between acute pancreatitis and Epstein-Barr virus infection by integrated bioinformatics analysis". Here is my detailed review:

Overall Assessment:

This manuscript presents a novel bioinformatics analysis exploring shared molecular mechanisms between acute pancreatitis (AP) and Epstein-Barr virus (EBV) infection. The study is well-designed, employs appropriate methodologies, and provides valuable insights into potential biomarkers and therapeutic targets. However, there are some concerns that need to be addressed before publication.

Strengths:

•Novel approach: The integrated analysis of miRNA-mRNA networks in AP and EBV infection is innovative and clinically relevant.

•Comprehensive methodology: The authors utilized multiple bioinformatics tools and databases, enhancing the robustness of their findings.

•Clinical implications: The identified miRNA-mRNA network and potential drug targets have clear translational potential.

•Well-structured: The manuscript is logically organized and clearly written.

Weaknesses:

•Limited experimental validation: The study relies heavily on in silico analysis without wet-lab confirmation of key findings.

•Sample size: Some datasets used have relatively small sample sizes, which may affect the generalizability of results.

•Lack of functional studies: The biological significance of identified miRNAs and mRNAs in AP and EBV infection is not experimentally demonstrated.

Detailed Comments:

Introduction:

•Well-written and provides adequate background.

•The rationale for the study is clearly stated.

Methods:

•Comprehensive and appropriate for the study objectives.

•The use of multiple datasets and various bioinformatics tools is commendable.

•Consider providing more details on the criteria for selecting the specific datasets used.

Results:

•The findings are presented clearly and logically.

•Figures are informative and well-designed.

•The identification of a miRNA-mRNA network comprising 5 miRNAs and 10 genes is a significant contribution.

•The potential role of HDAC inhibitors as therapeutic agents is an interesting finding.

Discussion:

•The authors provide a thorough interpretation of their results in the context of existing literature.

•The limitations of the study are appropriately acknowledged.

•Consider discussing the potential impact of tissue-specific differences, as some data are derived from peripheral blood rather than pancreatic tissue.

Conclusion:

•Concise and accurately reflects the main findings of the study.

References:

•Comprehensive and up-to-date.

Minor Issues:

•Some typographical errors need correction (e.g., "miRNAs" is sometimes written as "miR NAs").

•Ensure consistent formatting of gene symbols throughout the manuscript.

Recommendation:

This manuscript is recommended for publication after minor revisions. The authors should address the following points:

Discuss plans for experimental validation of key findings in future studies.

Provide a more detailed rationale for the selection of specific datasets used in the analysis.

Address the potential limitations of using peripheral blood data for understanding pancreatic pathology.

Correct minor typographical errors and ensure consistent formatting.

In conclusion, this study provides valuable insights into the shared molecular mechanisms between AP and EBV infection and has the potential to guide future research in this area. With the suggested revisions, it should make a significant contribution to the field.

**Reviewer #3:** I feel that this manuscript is very poor. I believe that it lacks proper research value and should not be accepted. The reasons are as follows:

1.In the introduction section, the author proposed that the occurrence of AP is related to EBV infection, and 90% of IM is caused by EBV virus. However, the relationship between AP and IM was not clearly explained. Moreover, the author did not clarify the research significance of TF-miRNA-mRNA.

2.There is a question, is the data source of GSE109220 from IM pathological specimens caused by EBV?

3.Dataset GSE42455 from AP rat models, which is inconsistent with the other three sets of data from humans, resulting in all subsequent analysis results having no scientific basis.

6. PLOS authors have the option to publish the peer review history of their article (what does this mean?). If published, this will include your full peer review and any attached files.

Reviewer #1: **Yes: **Shixiong Wei

Reviewer #2: No

Reviewer #3: No

---

## [Author Response · Author response to Decision Letter 0]

6 Sep 2024

Dear reviewers:

Thank you for your decision and constructive comments on our manuscript. We have carefully considered the comments and tried our best to address every one of them. According to your comments, revision notes, point-to-point, are given as follows:

Reviewer #1

Reviewer #1: The manuscript titled "Exploration of a miRNA-mRNA network shared between acute pancreatitis and Epstein-Barr virus infection by integrated bioinformatics analysis" presents an interesting study that delves into the shared molecular mechanisms between acute pancreatitis (AP) and Epstein-Barr virus (EBV) infection. The use of integrated bioinformatics analysis to identify miRNA-mRNA networks is commendable, and the study's findings could contribute to a deeper understanding of these pathological conditions. However, several areas need improvement to enhance the manuscript's clarity, rigor, and overall impact.

Firstly, the manuscript would benefit from a more detailed explanation of the methods used in the differential expression analysis and network construction. While the authors have provided a general overview, the specific criteria for selecting differentially expressed genes (DEGs) and miRNAs (DEMs) and the rationale behind these choices need to be more clearly articulated. Additionally, the process of intersecting DEGs and DEMs to construct the miRNA-mRNA network should be described with greater precision, including any assumptions made or thresholds applied.

Response:

We are so grateful for your kind question. Differential expression analysis is currently one of the mainstream analytical methods for finding disease-related molecules. The package DESeq2 provides methods to test for differential expression by use of negative binomial generalized linear models on un-normalized RNA-seq counts data. Limma is an R package for the analysis of gene expression data, especially the use of linear models for assessment of differential expression on normalized microarray data. DEGs or DEMs mean that there is a statistically significant difference in the expressions between two groups of samples. The choice of the specific cutoff value will generally be based on the experimental design and the objectives of the study. The cutoff value of fold change is usually set to 1.3 , 1.5 or 2 , with a corrected p-value < 0.05 , sometimes a p-value < 0.05 . In order to adequately identify more DEGs and DEMs in the present study, we set the cutoff value of fold change to 1.5, meaning the differential gene or miRNA for which the expression level of the disease group samples was up- or down-regulated by a factor of 1.5 compared with that of the control samples, and it is also necessary to satisfy the corrected p-value < 0.05 in the datasets GSE194331, GSE45918 and GSE109220. Moreover, a slight adjustment was made to the threshold criteria of significance level, setting it at a p-value < 0.05 within the dataset GSE42455. This was because only 11 DEMs were identified using the corrected p-value < 0.05 as the threshold.

The get_multimir() function of the R package multiMiR was employed to retrieve the DEMs target genes . The multiMiR contains a wide collection of validated and predicted miRNA–target interactions and their associations with drugs and diseases. It is composed of 14 external databases, which include 3 validated, 8 predicted, and 3 drug- or disease-related miRNA-target databases. In order to more accurately predict target genes of a given miRNA, we only considered validated miRNA-target interactions from three databases (miRecords, miRTarBase and TarBase). According to the detailed information provided by the multiMiR website (http://multimir.org/), there are 3045, 595913 and 644186 miRNA-target interactions records from miRecords, miRTarBase and TarBase, respectively. Given that the miRecords database contains approximately 3000 records, significantly fewer than miRTarBase and TarBase, and has not been updated since 2013, we opted to focus on miRNA-target gene interactions present in a minimum of two databases instead of requiring overlap across all three databases to ascertain the ultimate target genes.

Using the 8 overlapping DEMs as input, a total of 25, 1057, and 1055 validated miRNA-target interactions were found in miRecords, miRTarBase, and TarBase databases by multiMiR package, respectively (Supporting information, S6 Table). Then 960 target genes identified in at least two databases were further selected. A total of 10 overlapping genes were identified by taking intersections in 960 target genes and 111 DEGs. Finally, a miRNA-mRNA network of 11 miRNA–mRNA interaction pairs was constructed, which included 5 DEMs: miR-24-3p, miR-23b-3p, miR-130a-3p, miR-324-5p, miR-497-5p, and 10 DEGs: E2F2, HMGB2, LDHA, UBE2C, LMNB1, CEP55, AHNAK, AKT3, PLAG1, PLEKHA1.

Reviewer #1: The enrichment analysis results, while comprehensive, could be more effectively integrated into the narrative. Currently, the findings from the Gene Ontology (GO), Kyoto Encyclopedia of Genes and Genomes (KEGG), and disease ontology analyses are presented in a somewhat fragmented manner. It would be beneficial to discuss these results in the context of the broader biological implications and how they relate to the known pathophysiology of AP and EBV infection. Furthermore, the study would benefit from a more critical discussion of the limitations of enrichment analysis, particularly the potential for over-interpretation of results due to the inherent biases in the databases used.

Response:

Thank you very much for your comments and professional advice. Enrichment analysis helps researchers gain mechanistic insight into gene or miRNA lists generated from genome-scale experiments. In this study, at the threshold for adjusted p-value < 0.05, we performed enrichment analysis of genes and miRNAs using the R package clusterProfiler and the miRNA Enrichment Analysis and Annotation Tool (miEAA, version 2.1), respectively. We have further summarized the results of the enrichment analysis in the Results section as follows: Utilizing the 42 up-regulated DEGs as inputs, a collective of 34 biological process (BP) and 7 KEGG terms were significantly enriched, mainly comprising innate and adaptive immune response, interferon-gamma, complement and apoptotic signaling pathways. Using the 69 down-regulated DEGs as input, a total of 4 BP, 2 KEGG and 15 disease ontology terms were significantly enriched, which mainly included B cell proliferation, immune response, autoimmune diseases and tumors. Using the miEAA tool, a total of 234 subcategories were significantly enriched based on 8 DEMs. GO and KEGG subcategories mainly include cell cycle, nutrient metabolism and organ development. The diseases subcategories mainly encompass a wide range of solid and hematological tumors, neurological disorders and autoimmune diseases.

The discussion and limitations of the enrichment analysis results has also been supplemented in the third paragraph of the Discussion section as follows: Functional enrichment analysis showed that shared DEGs and DEMs between AP and EBV infection were involved in various biological processes and signaling pathways, such as innate and adaptive immune responses, autophagy and apoptosis signaling pathways. Innate immunity is the first line of antiviral defense and the activation of interferon signaling pathway is one of the major antiviral defense mechanisms of host cells after EBV infection . Consistent with our enrichment analysis results, a recent study also revealed that AP mice were enriched in biological processes related to immune system pathways and inflammatory responses . However, innate immunity is also a double-edged sword, as the induction of pro-inflammatory responses and the activation of programmed cell death . The pathophysiology of EBV-associated pancreatitis is unclear, but the pancreas is likely one of the target organs affected by the immune response. The unobstructed autophagy process is the basis for cells to maintain their metabolic functions, and the abnormal autophagy can cause autophagic vesicles to accumulation and lysosome inactivation, which is closely related to the occurrence of pancreatitis . Studies also find that EBV have evolved multiple strategies to interfere with autophagy to avoid destruction and promote their own replication and spread , . Increasing evidences indicate that EBV miRNAs and latent genes involved in manipulation of the cell apoptosis , . Mild acute pancreatitis was found to be associated with extensive apoptotic acinar cell death, whereas severe AP was found to involve extensive acinar cell necrosis but very little acinar cell apoptosis . Consistent with this, the results of our enrichment analysis found that apoptosis, but not necrosis, was significantly enriched. And the patients with EBV-associated AP were characterized by a favorable prognosis, without severe complications. However, since the dataset used for DEG and DEM identification are not from pancreatic tissue, it is still unclear which exact molecules and pathways are affected in the pancreas during EBV infection, and whether EBV can directly cause pancreatic cell damage and its potential molecular mechanisms still require further co-culture and in vivo studies.

Reviewer #1: The construction of the ceRNA and TF-miRNA-mRNA networks is an important aspect of the study. However, the manuscript would be strengthened by a more detailed discussion of the biological relevance of the identified networks. Specifically, the authors should provide more insight into the potential functional roles of the miRNAs and genes within these networks, as well as any known associations with AP and EBV infection. Additionally, the criteria for selecting the top 10 lncRNAs and circRNAs in the ceRNA network should be clarified, as this selection appears somewhat arbitrary.

Response:

We thank the reviewer for raising this question. ENCORI/starBase (Encyclopedia of RNA Interactomes) is an openly licensed and state-of-the-art platform to facilitate the integrative, interactive and versatile display of, as well as the comprehensive annotation and discovery of, RNA-RNA and protein-RNA interactions by deeply mining thousands of high-throughput sequencing data of RNA-RNA interactome, CLIP-seq and degradome sequencing. By inputting specific miRNA, miRNA-mRNA, miRNA-lncRNA, and miRNA-circRNA interactions predictions can be performed step by step. This highlights its prominence in ceRNA network construction studies , , . We have added detailed information on miRNA-target interactions predictions in the supplementary materials in order to present the predictions results comprehensively and objectively (S7 Table). After aggregation, however, there are thousands of interactions predictions. In order to simplify the interaction links for visualization, the interactions results were sorted by the ClipExpNum , which represented the number of supported experiments, with low number indicating weak interaction with the miRNA. Therefore, the corresponding top 10 lncRNAs and circRNAs candidates were selected for ceRNA network construction and visualization. 

In the meantime, the discussion on the biological relevance of the identified networks has been supplemented in the fourth paragraph of the Discussion section as follows: In order to showcase the upstream and downstream regulatory relationships of hub molecules, the ceRNA and TF-miRNA-mRNA networks were constructed. We found that lncRNA NEAT1 exhibited close relationships with five identified hub miRNAs in the ceRNA network. There are no reported associations between NEAT1 and EBV infection, but studies have shown that NEAT1 has opposite effects in limiting the replication of dengue and Hantaan viruses . Downregulation of NEAT1 relieves caerulein-induced cell apoptosis and inflammatory injury in pancreatic acinar cells through sponging miR-365a-3p . The NEAT1/miR-130/IRF1 axis in gastric cancer , NEAT1/miR-24-3p/LRG1 axis in thoracic aortic aneurysm , NEAT1/miR-324-5p/RAN axis in retinoblastoma and NEAT1/miR-497-5p/PIK3R1 axis in renal tubular oxidative injury have been widely studied. E2F1 acts as a master regulator in the TF-miRNA-mRNA network, which is a member of the E2F family of transcription factors and plays a crucial role in the control of cell cycle and action of tumor suppressor proteins. Studies have found that EBV-encoded proteins induce activation of E2F1 and S-phase entry, which may facilitate efficient lytic EBV replication in certain cell types , . A pan-cancer study indicated that KAT2A was found to cooperate with E2F1 and be recruited by E2F1 to the UBE2C promoter for elevating the expression of UBE2C by increasing the acetylation level of H3K9 to promote cell proliferation and the migration of cancer cells . And increased E2F1 expression via the E2F1/FTH1P3/miR-377-3p/LDHA axis promoted cell viability and glycolysis in laryngeal squamous cell carcinoma . In summary, although some studies have demonstrated the regulatory relationships in the ceRNA and TF-miRNA-mRNA networks, they still need to be further explored in the context of EBV infection and AP.

In addition, the potential functional roles of miRNAs and genes, as well as any known associations with AP and EBV infection were supplemented in the seventh and eighth paragraphs of the Discussion, respectively. Specifically, Su et al. reported that an miR-24-3p/MARCH3/NLRP3 axis mediated peritoneal macrophage M1 polarization and pyroptosis in AP . With the exception of miR-24-3p, there are no reports that the other four miRNAs are associated with AP or EBV infection; however, the studies , , , , , highlight the important role of these miRNAs in regulating inflammatory responses and provide a theoretical basis for further research in the context of AP or EBV infection. A sum of 10 shared genes were encapsulated in the miRNA-mRNA network. Among them, three genes (HMGB2 , LDHA , , and AKT3 ) were reported to be closely related to pancreatitis, and no gene was reported to be associated with EBV infection. However, these genes are directly or indirectly involved in the molecular events of AP and EBV infection, such as cell cycle regulation, the Ca2+ and Wnt signaling pathways and lymphocyte activation, and provide new directions for further research.

Reviewer #1: The immune cell infiltration analysis is a valuable addition to the study, but the interpretation of these results requires more depth. The manuscript should explore the implications of the observed differences in immune cell populations between AP and EBV-infected individuals and controls, and how these findings align with existing knowledge of immune responses in these conditions. Additionally, the correlation analysis between DEGs and immune cell populations could be expanded to include a discussion of potential mechanisms underlying these associations.

Response:

Thank you very much for your comments and professional advice. The discussion on the immune cell infiltration analysis has been supplemented in the fifth paragraph of the Discussion section. The results from immune cell infiltration analysis indicated that multiple immune cell populations were more abundant in AP samples than in controls, such as neutrophil and dendritic cells (DCs). Neutrophils are the first-line cells that migrate into the pancreatic inflammation site , and are central to the development of pancreatitis, mediating local tissue damage in the pancreas as well as distant organ damage and subsequent death , . Therapeutic interventions targeting neutrophils reduce tissue damage significantly and protect against the development of pancreatitis . DCs are known to be essential for pancreatitis recovery . Specifically, numbers of MHCII and CD11c dual-positive DCs increased 100-fold in pancreas of AP mice. Intrapancreatic DCs expressed higher levels of MHCII and CD86 and increased production of IL-6, membrane cofactor protein-1, and tumor necrosis factor-α. Depletion of DCs from mice with pancreatitis resulted in neutrophil infiltration and increased levels of systemic markers of inflammation, and died from acinar cell death within four days. A recent study distinguished six subsets of DCs in the pancreas and provided insights into their temporal behavior during pancreatitis . DCs

---

## [Decision Letter · Decision Letter 1]

12 Sep 2024

Exploration of a miRNA-mRNA network shared between acute pancreatitis and Epstein-Barr virus infection by integrated bioinformatics analysis

PONE-D-24-30500R1

Dear Dr. Wei,

We’re pleased to inform you that your manuscript has been judged scientifically suitable for publication and will be formally accepted for publication once it meets all outstanding technical requirements.

Kind regards,

Jinhui Liu

Academic Editor

PLOS ONE

Additional Editor Comments (optional):

I think this manuscript was well organized and it could be accepted.

Reviewers' comments:

Reviewer's Responses to Questions

**Comments to the Author**

1. If the authors have adequately addressed your comments raised in a previous round of review and you feel that this manuscript is now acceptable for publication, you may indicate that here to bypass the “Comments to the Author” section, enter your conflict of interest statement in the “Confidential to Editor” section, and submit your "Accept" recommendation.

Reviewer #1: All comments have been addressed

2. Is the manuscript technically sound, and do the data support the conclusions?

Reviewer #1: Yes

3. Has the statistical analysis been performed appropriately and rigorously? 

Reviewer #1: Yes

4. Have the authors made all data underlying the findings in their manuscript fully available?

Reviewer #1: Yes

5. Is the manuscript presented in an intelligible fashion and written in standard English?

Reviewer #1: Yes

6. Review Comments to the Author

Reviewer #1: (No Response)

7. PLOS authors have the option to publish the peer review history of their article (what does this mean?). If published, this will include your full peer review and any attached files.

Reviewer #1: **Yes: **Shixiong Wei

---

## [Editor Report · Acceptance letter]

18 Sep 2024

PONE-D-24-30500R1 

PLOS ONE

Dear Dr. Wei, 

I'm pleased to inform you that your manuscript has been deemed suitable for publication in PLOS ONE. Congratulations! Your manuscript is now being handed over to our production team.

Kind regards, 

on behalf of

Dr. Jinhui Liu 

Academic Editor

PLOS ONE